# Characterization of Marine-Surface-Dissolved Organic Matter via Amino Acid Enantiomers and Its Implications Based on Diel and Seasonal Observations

**Zhuo-Yi Zhu [1,2,\*], Ying-Chun Zhou [3] 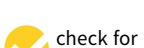, Wen-Chao Ma [2], Ying Wu [2], Ming Li [2], Su-Mei Liu [4], Xue-Wei Xu [1,5] and Meng Zhou [1]**

[1] School of Oceanography, Shanghai Jiao Tong University, Shanghai 200030, China; xuxw@sio.org.cn (X.-W.X.); meng.zhou@sjtu.edu.cn (M.Z.)

[2] State Key Laboratory of Estuarine and Coastal Research, East China Normal University, Shanghai 200241, China; 51173904021@stu.ecnu.edu.cn (W.-C.M.); wuying@sklec.ecnu.edu.cn (Y.W.); 51142601015@ecnu.cn (M.L.)

[3] Department of Statistics and Actuarial Sciences, East China Normal University, Shanghai 200241, China; yczhou@stat.ecnu.edu.cn

[4] Key Laboratory of Marine Chemistry Theory and Technology Ministry of Education, Ocean University of China, Qingdao 266100, China; sumeiliu@ouc.edu.cn

[5] Key Laboratory of Marine Ecosystem Dynamics, Ministry of Natural Resources & Second Institute of Oceanography, Ministry of Natural Resources, Hangzhou 310012, China

\* Correspondence: zhu.zhuoyi@sjtu.edu.cn

**Abstract:** Due to the essential roles of dissolved organic matter (DOM) in both microbiol food loop and marine carbon cycling, changes in marine DOM composition have an important impact on the marine ecosystem and carbon cycling. In October 2014 and June 2015, two field investigations for the DOM in the upper 200 m were conducted in the slope region of the northern South China Sea to characterize the DOM composition via amino acid enantiomers. In June, our sampling locations were under upwelling impact induced by an eddy-pair event, whereas in October there were no eddies. High-frequency sampling (a few hours interval) over 24 h reveals that the variability of the amino acid carbon yield (min. 0.2%) and the D/L alanine ratio (min. 0.02) is larger than its corresponding analytical and propagated errors, suggesting solid short-term changes for these two molecular-based indicators. Section samples from June showed a lower D/L alanine ratio (0.43 vs. 0.53) and a GABA mol% (1.0% vs. 1.6%) relative to the section samples from October, suggesting that DOM in June is more fresh (less degraded) compared to that in October. A higher serine mol% (19.5% vs. 13.2%) and lower D/L serine ratio (0.06 vs. 0.24) from the diel observation in June relative to October further indicates that phytoplankton, rather than bacteria, plays an more important role in DOM composition alternation. This is consistent with the higher phytoplankton biomass found in June, promoted by the eddy-pair.

**Keywords:** Amino acid; diel variability; dissolved organic matter; propagated error; South China Sea

## 1. Introduction

Differences in the composition of dissolved organic matter (DOM) among samples provide an indication of sample origins and reflect the refractory or labile nature of DOM [1,2]. Therefore, understanding the composition of DOM is an essential requirement in studies of carbon and nitrogen cycling, e.g., [3,4]. Amino acids (AAs) account for a small fraction of the bulk DOM, but they offer a powerful tool for determining the composition of organic matter and are widely used in contemporary organic matter studies, e.g., [1,5–8]. AAs are derived mainly from living organisms, and variations in both their total amounts and relative abundances are tightly coupled to the metabolism of organisms. Therefore, both the concentrations and relative abundances of AAs are dependent on the status or progress

of their metabolism (in the short term, such as within a day) and upon their source (in the longer term, such as seasonally).

In addition to the diel rhythm of photosynthesis (day and night) in phytoplankton, bacteria are periodically active in surface waters, even though their abundances are temporally similar [9]. The coupling of phytoplankton production, organic matter generation, and bacterial production in oligotrophic surface waters on a scale of hours has been proposed [10]. Such variations in microbial activity over a diel cycle have been repeatedly reported and confirmed, e.g., [11,12]. With respect to DOM, a labile fraction of bulk dissolved organic carbon (DOC) fluctuates over hours to days, as reviewed by Hansell (2013) [13]. In addition to the bulk DOC concentrations, the composition of DOM also shows temporal variability. The dissolved free AAs varied from 46 nM to 160 nM within 24 h at an inner Baltic Sea site [14]. Indeed, the DOM composition is frequently sampled and monitored in studies of diazotrophs and their fixation of nitrogen [15–17]. These studies have shown that by releasing dissolved organic nitrogen (DON), diazotrophs in oligotrophic waters affect DOM over very short time scales. However, despite frequent observations of dissolved free AAs and other free small molecules [18,19], total hydrolysable dissolved AAs (THDAAs) are rarely reported in diel observations of DOM. Though THDAAs and their relevant proxy variations over days to thousands of years are well revealed [20,21], the high-frequent variation degree of marine DOM within 1 day remains surprisingly unclear, which further interferes with our understanding of DOM composition over seasonal or even longer time scales.

Predominantly composed of combined AAs, THDAAs usually turn over much more slowly than dissolved free AAs or other small organic molecules. However, some THDAAs are also active and show changes in both field observations [21] and laboratory cultures [22], although the time scale (sampling interval) is usually longer than 1 day. The extent or degree of the THDAA changes within 1 day remains largely unclear. Such changes can be blurred due to their spatially heterogeneous distribution (in the case of field sampling) and/or by analytical errors, but should be attributable, in large part, to variations in microbial metabolism. It is also helpful to understand the temporal variability in THDAAs within 24 h, because such short-term changes allow us to more confidently clarify the changes in molecular indicators over longer timescales.

The northern slope of the South China Sea (SCS) is an oligotrophic open sea (Figure 1), usually characterized by nitrogen limitation [23]. In autumn and winter, vertical mixing in the upper water column is stronger, relative to spring and summer. Occasional eddy events further enhance the vertical gradient feature from season to season. The phytoplankton biomass remains low, but both the biomass and community structure can change a lot in response to the water column structure (e.g., occasional eddies) [24].

Within the framework of a national key research program, this study was part of multidisciplinary investigations that included comprehensive physical, chemical, and biological studies undertaken in the study area. With a clear background in the northern SCS, the questions in this work are: (1) what are the short-term changes in AA-based molecular indicators in the northern SCS within 1 day, and based on the achieved short-term changes? (2) what is the DOM composition difference in surface waters (<200 m) between June and October and its implications? To answer the questions, time-series observations and section observations for the upper 200 m were conducted in the northern slope area of the SCS in October 2014 and June 2015, respectively. In this work, we first quantify the propagated errors in our laboratory measurements and evaluate the degrees of variability of the AA-based molecular indicators over 24 h. Then, the seasonal DOM composition difference in the upper 200 m along the section is characterized. At last, the potential implications, including the DOM sources and degradation status, are addressed based on the AA-based molecular indicators.

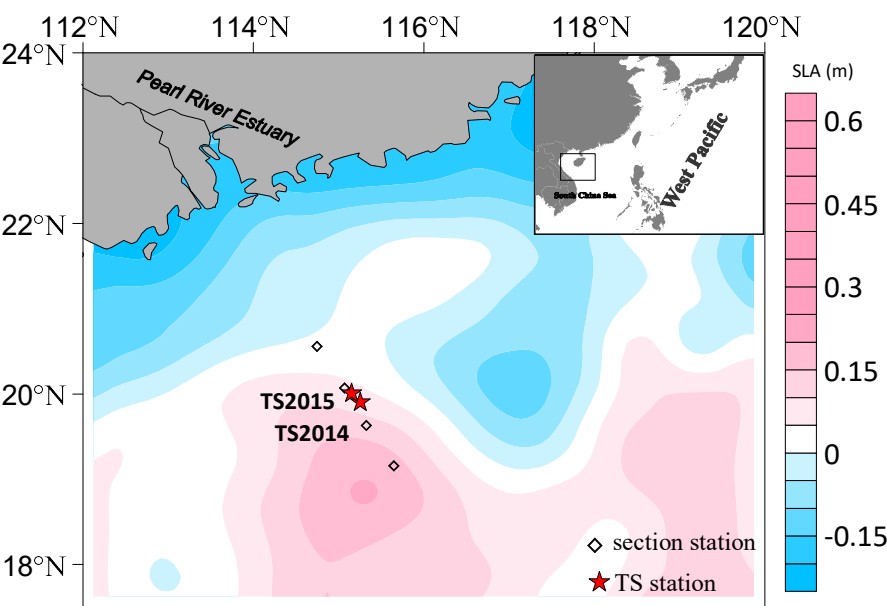

**Figure 1.** Study area and sampling stations in this study. The color contour indicates the sea level anomaly (m) on 15 June 2015. Sea level anomaly data downloaded from https://las.aviso.altimetry.fr/las/. An eddy-pair is very clear in June 2015, while no eddy was found in October 2014.

## 2. Materials and Methods

### 2.1. Background

The benefits of multidisciplinary investigations and the physical, chemical, and biological background of this work are basically revealed [25]. In June 2015, an eddy-pair was found during our diel observations, whereas in October 2014, no mesoscale process was detected (Figure 1) [26]. The observation sites were impacted by weak upwelling (Figure 1) [26]. Consequently, the surface phytoplankton biomass (0.12–0.48 µg L$^{-1}$) was higher in June 2015 than in October 2014, when the biomass was as low as 0.02 µg L$^{-1}$ [27]. In June 2015, the phytoplankton community was dominated by *Chaetoceros* and *Thalassionema*, both of which are diatoms [27]. Although their abundance remained unclear in previous work, when the bacterial diversity was studied in October 2014 the community was dominated by heterotrophs, and the abundance of autotrophic bacteria was rather low [28].

### 2.2. Field Sampling

Two cruises were undertaken in R/V *Nan Feng* on the northern slope region of the SCS during 10–30 October 2014 and 10–30 June 2015. The temperature (T), salinity (S), fluorescence, and photosynthetically active radiation (PAR; in June 2015 only) were measured with a CTD (Seabird, USA). Discrete water samples were collected in depth profiles (top 200 m) along a section that covers the SCS northern slope area (Figure 1). In addition, time-series stations (TS2014 in 2014 and TS2015 in 2015) were also observed. For the time-series stations, sampling began at 06:00 and lasted for 1 day at TS2014. At TS2015, sampling began at 12:35 and also lasted for 1 day. In both cruises, the time-series observations were carried out 1 week later when compared to the corresponding section observations. Up to five profile samplings were conducted during the time-series stations in each of the two seasons. The sampling interval was approximately 6 h.

Immediately after collection, the seawater samples for the analysis of AAs and DOC were filtered with a clean nylon membrane (pore size 0.45 µm) and a syringe, and frozen (−20 °C) before analysis in the laboratory. Water samples for the analysis of dissolved nutrients were first filtered through acid-cleaned acetate cellulose filters (pore size 0.45 µM), treated with HgCl$_2$, and stored at 4 °C in the dark before analysis. Nutrient samples were not collected during the last two samplings at TS2014.

At TS2014, water samples were further collected for the detection of bacterial abundance at 15:00 and 21:00 on 26 October (i.e., 3 h after the molecular indicators were sampled), whereas in late spring (June) at TS2015, the water samples for bacterial abundance were collected at the same time as the samples used to analyze the molecular indicators (one sampling point was missed: the 2nd sampling point). A volume of 47.5 mL of seawater was collected and fixed with 2.5 mL of formaldehyde for 15 min. The samples were stored at 2–8 °C and analyzed in the laboratory.

*2.3. Laboratory Measurements*

All the measured AAs (including both L- and D-enantiomers) and their abbreviations are shown in Table 1. The THDAAs were analyzed with a previously described method [29]. Briefly, the water samples were first hydrolyzed in sealed ampoules with 6 M HCl at 110 °C under $N_2$. For samples with elevated nitrate levels, ascorbic acid was also added [30]. After hydrolysis, the samples were neutralized and the pH adjusted to 8.5. After precolumn derivatization with o-phthaldiadehyde (OPA) and *N*-isobutyryl-L/D cysteine, the AA enantiomers were separated and measured in the hydrolysates with high-performance liquid chromatography (HPLC; 1200 series Agilent, USA), with a fluorescence detector. During hydrolysis, asparagine (Asn) and glutamine (Gln) were deaminated to aspartic acid and glutamic acid, respectively. Therefore, "Asx" represents Asn plus aspartic acid (Asp), and "Glx" represents glutamine plus glutamic acid (Glu) (Table 1). The DOC concentrations were measured with a total organic carbon (TOC) analyzer (TOC-L$_{CPH}$, Shimazu, Japan). Ammonium was measured manually by the sodium hypobromite oxidation method, with an analytical precision of 0.04 μM. Concentrations of the other four nutrients were determined with an auto-analyzer (AA3: SEAL Analytical, Mequon, WI, USA), with precisions for nitrate, nitrite, dissolved inorganic phosphorus (DIP, $PO_4^{3-}$), and silicate ($SiO_3^{2-}$) of 0.01, 0.003, 0.005, and 0.02 μM, respectively. Milli-Q water was used in all laboratory processing, and the chemicals were purchased from Sigma-Aldrich (Missouri, USA) or Merck (USA) and were of HPLC grade or above.

**Table 1.** Measured amino acids and abbreviations used in this study (L- and D-enantiomers are not listed).

| Name | Abbreviation |
| --- | --- |
| Total hydrolysable dissolved amino acids | THDAAs |
| Alanine | Ala |
| Arginine | Arg |
| Asparagine Aspartic acid | Asx |
| Glutamine Glutamic acid | Glx |
| Glycine | Gly |
| Isoleucine | Ile |
| Leucine | Leu |
| Lysine | Lys |
| Methionine | Met |
| Phenylalanine | Phe |
| Serine | Ser |
| Threonine | Thr |
| Tyrosine | Tyr |
| Valine | Val |
| γ–aminobutyric acid | GABA |
| 2–aminobutyric acid | Aba |

To determine bacterial abundance, 1 mL 4′,6-diamidino-2-phenylindole solution (DAPI; 10 μg mL$^{-1}$) was added to 30 mL of each sample, and the mixtures were incubated in the dark for 5–10 min. Each sample was then filtered through 25 mm, and 0.2 μm black Whatman® Nuclepore™ Polycarbonate Filters were mounted on pre-wetted

0.8 Nuclepore Membra-fil® cellulosic filters [31]. The filters were then set onto glass slides with a drop of Leica type A immersion oil. The bacterial abundances were determined with an epifluorescence microscope (Leica, DM5000B).

### 2.4. The AA-Based Molecular Indicators and Propagated Error Quantification

Among the AA-based molecular indicators, AA carbon yield is the AA carbon (in nM) divided by bulk DOC (in nM). AA% is the mol percentage of a given AA in THDAA. The degradation index (DI) is a measure that tells the overall degradation status of organic matter. The DI, and its calculation equation, was first proposed by Dauwe and Middelburg (1998) [32] for use in particulate and sediment studies. Later, it was applied to DOM with a revised constant derived from updated DOM samples [21]. In this work we calculated DI following the method of Dauwe and Middelburg (1998) [32], using the revised coefficients derived from DOM sample pools [21,33].

To check the errors in the measurements of the AA enantiomers, a water sample (10L) obtained at a depth of 5 m near TS2014 was sealed and stored for 3 years at room temperature in the dark to ensure it reached a stable and homogeneous state. Three subsamples were then taken from the 10 L sample. The concentrations of the THDAAs in the subsamples were determined, using the sampling and laboratory procedures described above. The standard deviation of the three subsamples was regarded as the error of the primary measurement of DCAAs.

To quantify the errors in measuring DOC, we used the deep-sea water standard (CRM-batch 9) from the Consensus Reference Materials project led by Prof. Dennis Hansell, at the University of Miami (FL, USA). Up to eight samples were analyzed. Similarly, the standard deviation of the eight DOC values was regarded as the error of the primary measurement of DOC.

When deriving the AA-based molecular indicators (D/L ratio, Gly%, percentage γ-aminobutyric acid (GABA%), AA carbon yield, and DI) from the primary measurements (i.e., individual AA concentrations and/or DOC concentration), the errors within the primary AA and/or DOC measurements were propagated to these molecular indicators. To obtain the equations for the propagation of uncertainties (as standard deviations), a statistical inference was conducted for each molecular indicator. The equations used to calculate the propagated errors are provided in the Supplementary Materials.

### 3. Results

#### 3.1. Time-Series Observations

#### 3.1.1. Physical Parameters, Nutrients, and Bacteria

In the surface waters, the temperature and salinity for October 2014 (June 2015) were 27.3 °C (30.1 °C) and 33.7 (32.8), respectively. There were very limited fluctuations in the surface-water temperature (maximum − minimum = 0.1 °C) or salinity (maximum − minimum = 0.1) over the diel observations in October 2014, whereas the fluctuations in both temperature (maximum − minimum = 0.4 °C) and salinity (maximum − minimum = 1.2) were greater in June 2015. In October 2014, a fluorescence peak was identified at 66 m depth at noon, decreasing both upwards and downwards. In June 2015, the fluorescence peak occurred at a depth of 40–54 m, and the depth at which the PAR value was 1% of that at the surface (i.e., the depth of the euphotic zone) was 56 m at that time.

The nutrient levels differed between June and October. In the surface waters, dissolved inorganic nitrogen (DIN) was slightly higher in June than in October (0.3 μM and 0.1 μM, respectively; Table 2). The DIP concentration was higher in October 2014 (mean 2.1 μM at the surface) than in June 2015, when it ranged from 0.03 μM to below the detection limit. A similar seasonality was observed at 200 m depth, with higher DIN and lower DIP in June but lower DIN and higher DIP in October (Table 2). Silicate was also clearly higher in June (2.0 μM) than in October (0.03 μM) in the surface waters and at a depth of 200 m (20 μM and 0.8 μM, respectively (Table 2). The DIN/DIP ratio in the surface waters was very low (0.05) in October and higher (7.6) in June. At 200 m depth, the DIN/DIP ratio was 0.8 in

October and 18 in June (Table 2). Over the entire upper 200 m water column (i.e., extending between 0 and 200 m), the DIN/DIP ratio was 0.48 and 16.66 over our diel observation period in October and June, respectively.

**Table 2.** Inorganic nutrients (μM) and dissolved inorganic nitrogen (DIN)/dissolved inorganic phosphorus (DIP) ratios recorded in the diel observations in this study [a].

|  |  |  | Ammonium | Nitrite | Nitrate | DIP | Silicate | DIN | DIN/DIP |
|---|---|---|---|---|---|---|---|---|---|
| 201410 [b] | 5 m | Mean | **0.1** | **bdl** [a] | **bdl** [a] | **2.1** | **0.03** | **0.1** | **0.05** |
|  |  | sd [c] | 0.02 |  |  | 0.05 | 0.02 | 0.02 | 0.01 |
|  |  | Min | 0.08 |  |  | 2.0 | 0.01 | 0.08 | 0.04 |
|  |  | Max | 0.1 |  |  | 2.1 | 0.05 | 0.1 | 0.06 |
|  | 200 m | Mean | **0.06** | **0.01** | **11.2** | **14** | **0.8** | **11** | **0.8** |
|  |  | sd [c] | 0.01 | 0 | 1.3 | 2.7 | 0.1 | 1.3 | 0.05 |
|  |  | Min | 0.05 | 0.01 | 10 | 12 | 0.7 | 10 | 0.7 |
|  |  | Max | 0.07 | 0.01 | 13 | 17 | 1.0 | 13 | 0.8 |
| 201506 | 5 m | Mean | **0.15** | **0.01** | **0.1** | **0.04** | **2.0** | **0.3** | **7.6** |
|  |  | sd [c] | 0.03 | 0.01 | 0.2 | 0.01 | 0.3 | 0.2 | 4.3 |
|  |  | Min | 0.1 | bdl [a] | bdl [a] | 0.03 | 1.6 | 0.1 | 4 |
|  |  | Max | 0.18 | 0.02 | 0.4 | 0.05 | 2.4 | 0.6 | 12 |
|  | 200 m | Mean | **0.2** | **0.02** | **14** | **0.8** | **20** | **15** | **18** |
|  |  | sd [c] | 0.2 | 0.01 | 2.1 | 0.2 | 5 | 2 | 2 |
|  |  | Min | 0.1 | 0.01 | 11 | 0.55 | 13 | 12 | 17 |
|  |  | Max | 0.5 | 0.04 | 17 | 1.0 | 26 | 17 | 22 |

[a] bdl: below detection limit; [b] Two sampling times are missing for October 2014; [c] sd, standard deviation.

In the surface waters, the bacterial abundance in October was 10 times that in June, with average values of $2.3 \times 10^8$ cell $L^{-1}$ and $2.3 \times 10^7$ cell $L^{-1}$, respectively (Table 3). The bacterial abundance decreased vertically in October; at 200 m depth, it was only half or one quarter that in the surface waters. In June, when the bacterial abundance in the surface waters remained low, it was similar from the surface to 200 m depth (or even increased).

**Table 3.** Background view of the two diel observations in this study.

| Month | Eddy Effect [a] | Chlorophyll a [b] μg $L^{-1}$ | DIN/DIP Surface | DIN/DIP Bottom | Bacteria $\times 10^7$ Cell $L^{-1}$ | Short Remarks |
|---|---|---|---|---|---|---|
| October | no | 0.02 | 0.05 | 0.8 | 23 | strong nitrogen limit throughout top 200 m; low chla/high bacteria |
| June | weak upwelling | 0.3 (0.12–0.48) | 7.6 | 18 | 2.3 | weak (or no) nitrogen limit over top 200 m, high chla/low bacteria |

[a] cited from Chen et al. [26]; [b] cited from Zhang et al. [27]; the key algal groups in both months were diatoms.

### 3.1.2. Temporal Variability of DOM over 24 h

The levels of DOC in surface waters were similar in October and June (mean: 83 μM), but their variability over 24 h differed. In October, DOC ranged between 80 μM and 86 μM, whereas in June, it ranged between 75 μM and 87 μM (Table 4). The ranges (6 μM in October and 12 μM in June) and the diel variability (standard deviations: 3 μM for October and 5 μM for June; Table 4) indicate that DOC varied more weakly in October than in June, and both values are clearly larger than our measurement precision (1 μM; Table S2). THDAAs in surface waters averaged $366 \pm 37$ nM in October, much lower than in June ($584 \pm 57$ nM; Table 4). Similarly, the AA carbon yield was also lower in October ($1.6\% \pm 0.2\%$) than in June ($2.3\% \pm 0.3\%$). The diel variability in surface THDAAs and the AA carbon yield were smaller in October (37 nM and 0.2%, respectively) than in June (57 nM and 0.3%, respectively), as those of DOC (Table 4).

The concentrations of, and variability in, the remaining molecular indicators are also shown in Table 4. In October, surface waters were characterized by lower Gly% ($30\% \pm 2\%$), lower Thr% ($3.7\% \pm 0.4\%$), lower Ser% ($14\% \pm 1.1\%$), higher D/L Ala ($0.34 \pm 0.04$), and

lower DI ($-0.52 \pm 0.28$) than the corresponding values in June (Gly%, 33% $\pm$ 2%; Thr%, 4.6% $\pm$ 1%; Ser%, 18.9% $\pm$ 0.8%; D/L Ala, 0.24 $\pm$ 0.03; DI, $-1.17 \pm 0.23$) (Table 4). The differences between these mean values (3% = 33% $-$ 30% for Gly%; 0.1 = 0.34 $-$ 0.24 for D/L Ala; 0.65 = $-0.52 - (-1.17)$ for DI) were all larger than the corresponding maximum propagated errors (0.5% for Gly%, 0.02 for D/L Ala, 0.2 for DI; Table S3), so the differences were pronounced. Although there was also a mathematical difference for GABA% (mean value: 1.1% in October and 0.8% in June in surface waters; Table 4), the resulting difference (0.3% = 1.1% $-$ 0.8%) was smaller than the maximum propagated error (0.4%; Table S3).

**Table 4.** Mean dissolved organic carbon (DOC) and amino-acid-based molecular indicator values, diel variations (as standard deviations (sd*), in bold font), and ranges (minimum–maximum) in the surface waters and at 200 m depth during the time-series observation in the South China Sea slope region.

|  |  |  | DOC | THDAAs | AA Carbon Yield | Gly | Thr | Ser | GABA | D/L Ala | DI |
|---|---|---|---|---|---|---|---|---|---|---|---|
|  |  |  | µM | nM | % | % | % | % | % |  |  |
| October 2014 | 5 m | mean | 83 | 366 | 1.6 | 30 | 3.7 | 14.0 | 1.1 | 0.34 | −0.52 |
|  |  | **sd*** | **3** | **37** | **0.2** | **2** | **0.4** | **1.1** | **0.6** | **0.04** | **0.28** |
|  |  | min | 80 | 341 | 1.4 | 28 | 3.0 | 12.3 | 0.5 | 0.30 | −0.97 |
|  |  | max | 86 | 428 | 1.9 | 33 | 4.0 | 15.1 | 1.9 | 0.40 | −0.20 |
|  | 200 m | mean | 68 | 292 | 1.5 | 32 | 3.7 | 13.1 | 1.0 | 0.30 | −0.93 |
|  |  | **sd*** | **3** | **69** | **0.4** | **2** | **1.0** | **0.9** | **0.5** | **0.13** | **0.18** |
|  |  | min | 65 | 174 | 0.9 | 30 | 2.9 | 11.7 | 0.3 | 0.18 | −1.21 |
|  |  | max | 71 | 341 | 1.8 | 34 | 5.3 | 14.2 | 1.5 | 0.50 | −0.75 |
| June 2015 | 5 m | mean | 83 | 584 | 2.3 | 33 | 4.6 | 18.9 | 0.8 | 0.24 | −1.17 |
|  |  | **sd*** | **5** | **57** | **0.3** | **2** | **1.0** | **0.8** | **0.2** | **0.03** | **0.23** |
|  |  | min | 75 | 507 | 1.8 | 31 | 3.1 | 18.1 | 0.5 | 0.21 | −1.49 |
|  |  | max | 87 | 647 | 2.5 | 34 | 5.5 | 20.2 | 1.0 | 0.27 | −0.99 |
|  | 200 m | mean | 50 | 351 | 2.3 | 33 | 3.7 | 19.1 | 0.7 | 0.27 | −1.20 |
|  |  | **sd*** | **2** | **76** | **0.4** | **1** | **0.5** | **0.8** | **0.3** | **0.02** | **0.27** |
|  |  | min | 47 | 232 | 1.7 | 32 | 3.0 | 18.2 | 0.4 | 0.25 | −1.51 |
|  |  | max | 52 | 436 | 2.8 | 35 | 4.3 | 19.9 | 0.9 | 0.30 | −0.76 |

The concentrations and diel variability at water depths of 200 m are also shown in Table 4. Briefly, DOC, AAs, and molecular indicators were lower at 200 m than in the surface waters (excluding Gly%) in both months. In June, Gly% and DI remained similar at 5 m and at 200 m depth, and D/L Ala was slightly higher at 200 m (0.27) than in the surface waters (0.24; Table 4).

### 3.1.3. Temporal Variations in DOM

In October, the peak DOC (86 µM) in surface waters occurred at dusk (18:00), and the lowest concentration (80 µM) occurred at midnight (24:00). However, there was no comparable trend in June, when the peak DOC concentration (87 µM) occurred at dawn (06:00) and the lowest concentration (75 µM) occurred at dusk (18:00) (Figure 2a).

The diel THDAA pattern seems to have been decoupled from the variation in DOC. In October, THDAAs in the surface waters remained almost unchanged from 06:00 to midnight (24:00), when the concentration was slightly increased (Figure 2b). In June, the THDAA pattern did not mirror the corresponding DOC pattern either, but dropped steadily from 12:00 till the next dawn (06:00) and then increased again to 12:00 on the following day (Figure 2b). The AA carbon yield pattern was very similar to that of THDAAs (Figure 2c).

Unlike the decoupling of THDAAs and DOC, the variations in the molecular indicators seemed to reflect some variations in DOC in October. At 18:00, when DOC in the surface water was highest, Gly%, D/L Ala, and DI in the surface waters were also at their maxima over the observation period (Figure 2d–f). Moreover, at night (24:00) and dawn (06:00), when DOC was low, Gly%, D/L Ala, and DI were also relatively low (Figure 2d–f). However, in June, there was no comparable coupling between DOC and the molecular indicators; instead, the indicators showed the reverse pattern relative to DOC (e.g., Gly% was elevated at 18:00 when DOC was low, and remained depleted at 06:00 when DOC was high; Figure 2d).

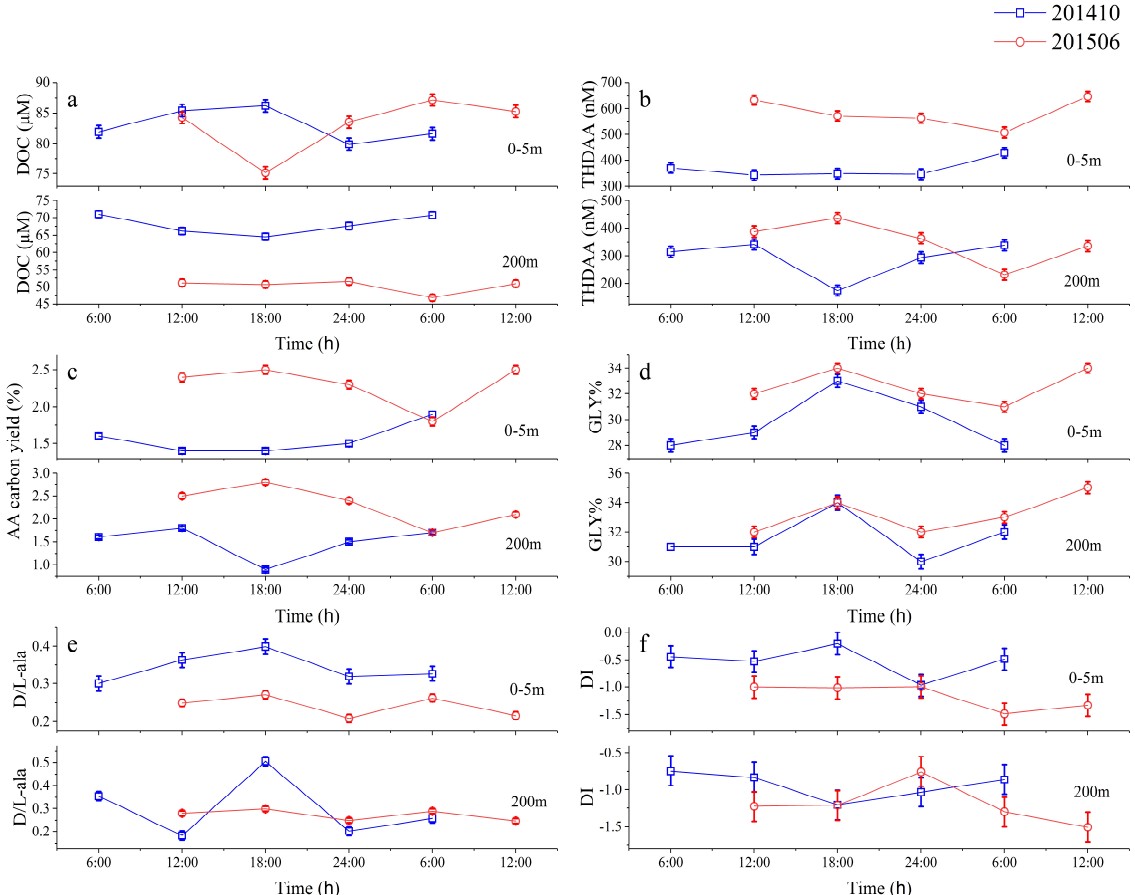

**Figure 2.** Variations in (**a**) DOC, (**b**) THDAA, (**c**) AA carbon yield, (**d**) Gly%, (**e**) D/L-Ala and (**f**) DI with time in the SCS slope region; error bars are the corresponding propagated errors in October and June.

### 3.2. Section Observations

In surface waters, vertical mixing was deeper in October than in June, as indicated by the density distributions [quantified as σ in kg/m$^3$, = 1000 × (seawater density–1.0)]. Waters with density 22.5 kg/m$^3$ were almost universally distributed within the 50 m top waters in October 2014, whereas in June 2015 such waters (22.5 kg/m$^3$) were instead distributed in a much shallower depth (~25 m; Figure 3). In the bottom waters beneath 100 m, a clear upwelling could be identified in June 2015 as indicated by the upwelled waters with density 25 kg/m$^3$, whereas in October 2014 stratification prevailed in the depth > 100 m. The upwelling in June relative to October is also indicated by the upper boundary of waters with density 25 kg/m$^3$, which in October 2014 was deeper when compared to that in June 2015 (Figure 3). A clear deep chlorophyll maximum (DCM) belt can be found along the section in both October and June (Figure 3). The DCM belt roughly followed the waters with density 23–23.5 kg/m$^3$ in both months (Figure 3), occurring at a depth between 50 to 100 m. The fluorescence signal was more intensive in June relative to that in October.

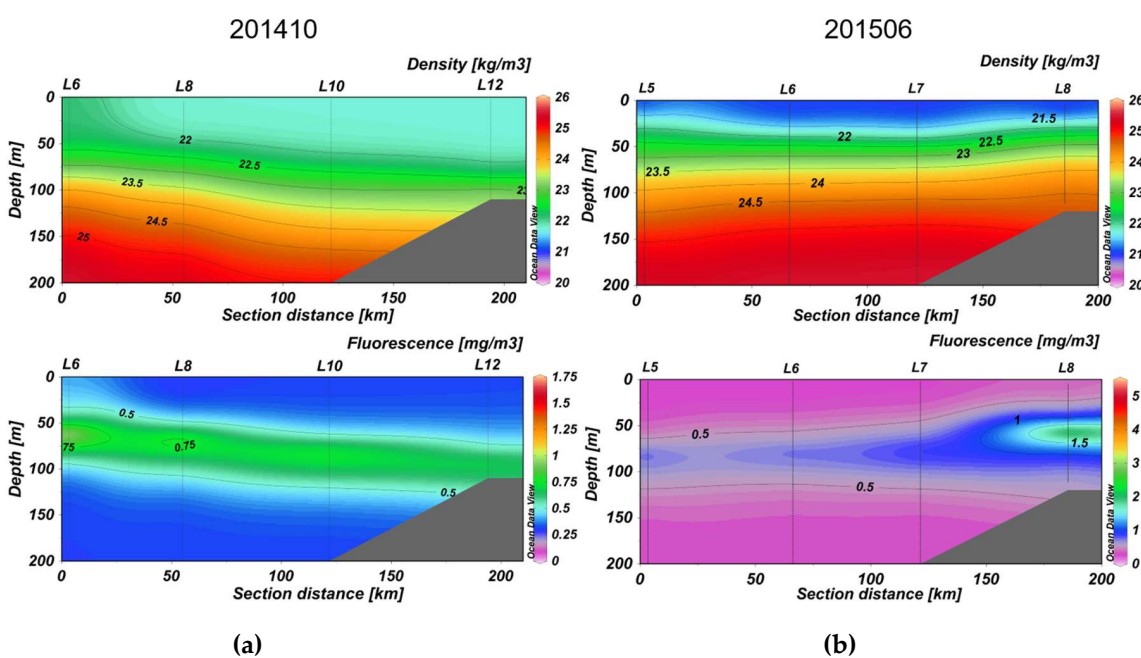

**Figure 3.** Density and fluorescence distribution patterns in the section for (**a**) October 2014 and (**b**) June 2015.

Overall, THDAA was higher in June relative to October. In October, THDAA ranged from 181 nM to 317 nM, with an average value of 252 nM, whereas in June the mean value was 452 nM (ranging from 257 to 596 nM) (Table 5). In June, the elevated THDAA values (e.g., >452 nM) basically appeared in the upper 100 m waters. The D/L Ala ratio in June was lower (0.43) when compared to that in October (0.53) (Table 5). In October, elevated D/L Ala ratios roughly overlapped with the DCM zone, but in June such overlap pattern disappeared. Though the discrete samples cannot ideally cover the DCM belt, as revealed by the CTD probe, the D/L Ala ratio was low (e.g., <0.4) on the DCM belt zone in June (Figures 3 and 4). Gly% and GABA% in June were also lower than in October (Table 5). As shown in Table 5, Gly% (GABA%) was 6% (0.6%) lower in June relative to October, judged by the mean values. Spatially, both Gly% and GABA% indicate a good DOM composition vertical mixing in June, whereas in October the DOM composition showed a horizontally laminated distribution feature (Figure 4). As for DI values, a higher DI was observed in June (mean: 0.49) relative to October (mean: 0.20) (Table 5).

**Table 5.** THDAA and key AA proxies in the upper 200 m as revealed by the section observations in this study.

|  | THDAA(nM) | D/L Ala | Gly% | GABA% | DI |
|---|---|---|---|---|---|
| October 2014 | 252 (181–317) | 0.53 (0.44–0.72) | 30 (26–35) | 1.6 (1.1–2.5) | 0.20 (−1.02–2.24) |
| June 2015 | 452 (257–596) | 0.43 (0.34–0.64) | 24 (17–31) | 1.0 (0.4–1.9) | 0.49 (−1.02–2.40) |

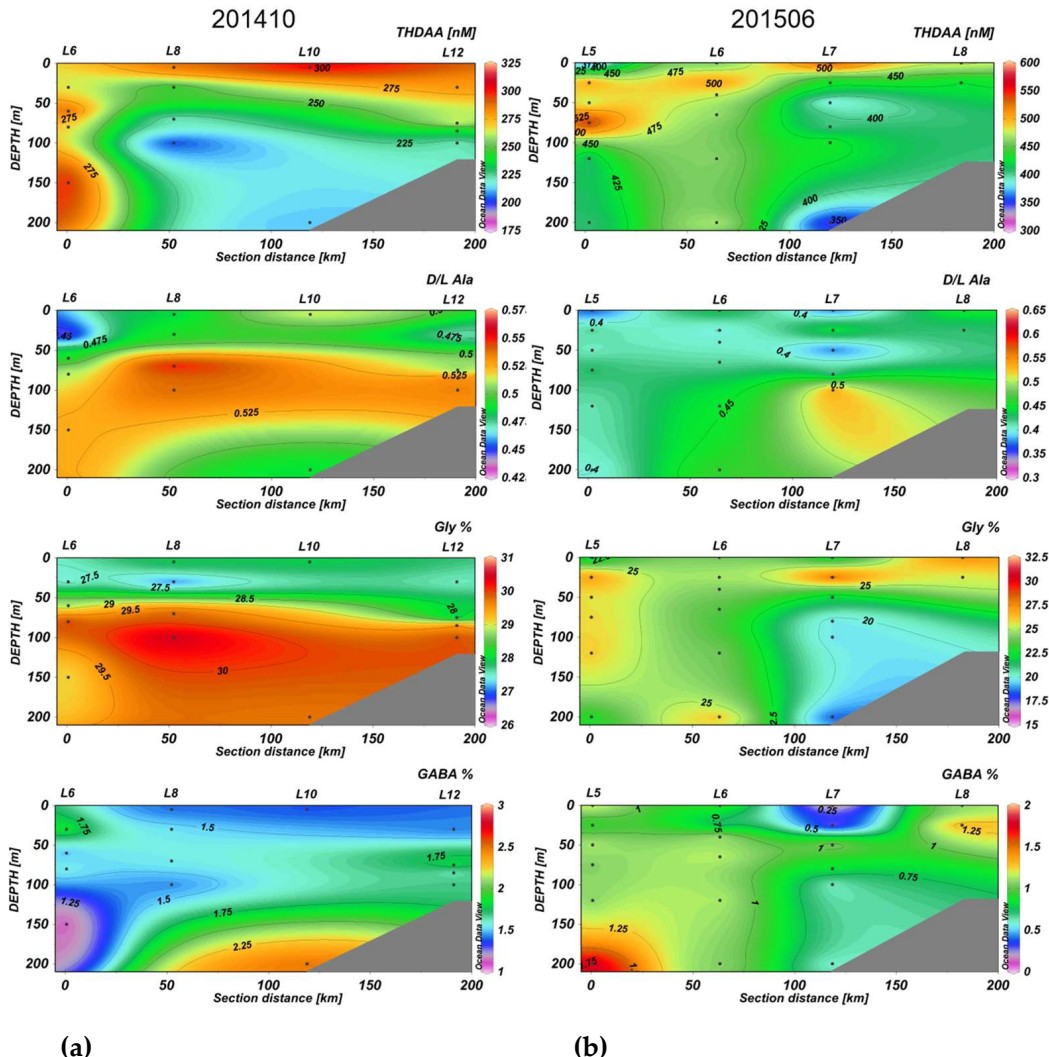

**Figure 4.** Total hydrolysable dissolved amino acid (THDAA) and key amino acid (AA) proxies distribution patterns in the section for (**a**) October 2014 and (**b**) June 2015.

## 4. Discussion

### 4.1. Solid Short-Term Changes of AA-Based Molecular Indicators

Overall, our results show that the propagated error played a minor role in most AA-based molecular indicators (Table S3). Though the propagated error for DI can be very high, this is not strange, as the DI value itself can be very close to zero [20,33,34]. Instead, we suggest a careful consideration of GABA% (Table S3). The absolute value of the propagated error for GABA% was of the order of 0.3–0.4%, which was similar to, or even larger than, the GABA% value itself, which was 0.3–2.0% in this study. Therefore, in studies using GABA% as a molecular indicator of surface DOM, this error should be considered. In deep waters, however, the impact of the propagated error on GABA% should be minor because GABA% increases with depth and can be >15% at water depths of >800 m [20].

The precision of the DOC and THDAA measurements was 1 μM and 20 nM, respectively, far lower than the corresponding short-term changes over the 24 h observation period (Figure 2a,b). For molecular indicators, a comparison of the short-term changes (expressed as standard deviations) and the corresponding seasonal propagated errors indicated that short-term changes were significant for AA carbon yield, Gly%, and D/L Ala ratio, whereas the other two molecular indicators were not (e.g., GABA%; Figure 5). The

variability (as a standard deviation) in GABA% over 24 h was similar to, or even smaller than, the propagated error (Figure 5).

The bulk DOC and THDAA changes over 24 h were clearly greater in June 2015 than in October 2014 (Figure 2a,b): for surface waters in June 2015, the standard deviations were 5 μM (DOC) and 57 nM (THDAAs), whereas in October 2014, they dropped to 3 μM (DOC) and 37 nM (THDAAs) (Table 4). The short-term changes in THDAAs (maximum–minimum differences of 87 nM for October and 140 nM for June in surface waters) were similar to the diel changes in DFAAs (maximum–minimum difference 114 nM) in the Baltic Sea [14].

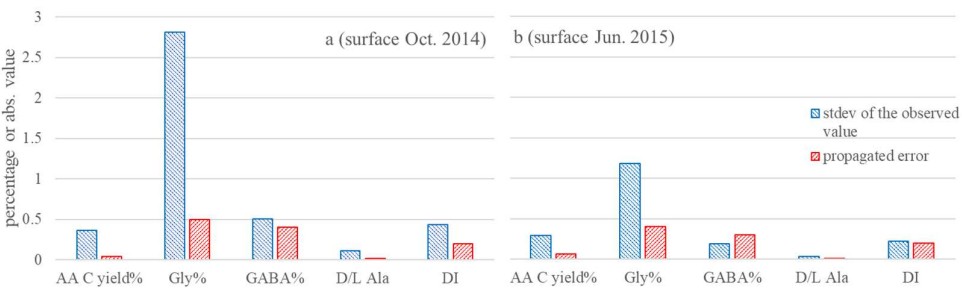

**Figure 5.** Comparison of standard deviations of the observed values over 24 h in surface water and the corresponding seasonal mean propagated errors. The pattern was very similar for the water at 200 m depth, and is therefore not shown.

In addition to their heterogeneous distribution, the short-term changes in DOM over 24 h above the euphotic zone may be related to light-driven autotrophic contributions and related heterotrophic processes [13], whereas the short-term changes below the euphotic zone (i.e., 200 m in this study) may be attributable to either the heterogeneous distribution of microbes (and hence the DOM concerned) or the contribution of swimmers (vertically migrating zooplankton and fish). It has been suggested that the vertical migration of mesopelagic fish (and their excretions) plays a role in the diel variability in DOM (DOC and carbohydrates) during our sampling period [35]. The bulk DOC concentration and AAs showed a gradual decrease from the surface to deeper layers in June, but the statistical analysis of the respective diel observation data set—namely in June and October, respectively—indicates there are no seasonal relations between water masses (temperature, salinity, and density) and AA-based molecular indicators. Statistically, the bacterial abundance in June showed a negative relation with DOC, suggesting the potential influence of bacteria on DOM short-term changes. However, the relation is very weak (r = −0.55, respectively; figure not shown). Bacteria are periodically active in surface waters, even though their abundances are temporally similar [9]. Hence, it is reasonable that bacteria are playing an important role in constraining the short-term changes of DOM, but more sensitive biological proxies (e.g., ATP, RNA, and sequencing) are needed to quantify the biological role in DOM short-term changes. The mechanism of the short-term changes in DOM composition need further study.

The AA carbon yield, D/L ratio, DI, and nonprotein AA% have been intensively discussed previously, and have shown different patterns of change during DOM diagenesis [21,36–38]. The AA carbon yield and the D/L ratio are most sensitive for tracking changes during the early stage of diagenesis (days to weeks) [20,36,38]. DI appears to be most effective during the intermediate stage of diagenesis (annual to decadal timescales), and nonprotein AA% (e.g., GABA% in this study) is the best indicator in the later stages of DOM diagenesis (e.g., >decades) [20]. Our results support these findings (Figure 5) in that the daily changes in the AA carbon yield were apparent. The diel changes in D/L Ala were similar to (or seasonally larger than) the propagated error. Neither DI nor GABA% showed significant short-term changes compared with the propagated error.

### 4.2. Seasonal DOM Composition Difference and Its Implications

In the northern slope region of the SCS, Gly% in DOM is comparable to DOM samples from other regions in the northern SCS, like the offshore of the Pearl River Estuary [39], and from oligotrophic surface oceanic waters [37] (Figure 6). Diatom frustules and bacterial cell membrane macromolecules are the two DOM sources that are relatively enriched in Gly in the open sea [6]. In addition to Gly, bacterial cell membrane macromolecules contain a few other D-form AAs, whereas diatom frustules are enriched in L-form Ser and Thr [40,41]. During our diel observations, the elevated Ser% (mainly contributed by L-Ser) was as high as 10–23%, which was much higher than at BATS or HOT (3–12%) or in other oceanic DOM (Figure 6; or see Table 4 in Jørgensen et al., 2014) [37]. However, our observed value (Ser%, 10–23%; Figure 6) was very similar to previously reported Ser% (9–21%) in the northern SCS (e.g., offshore waters from the Peal River Estuary) [39], suggesting that the elevated Ser% in the surface of the northern SCS DOM was a reproducible feature.

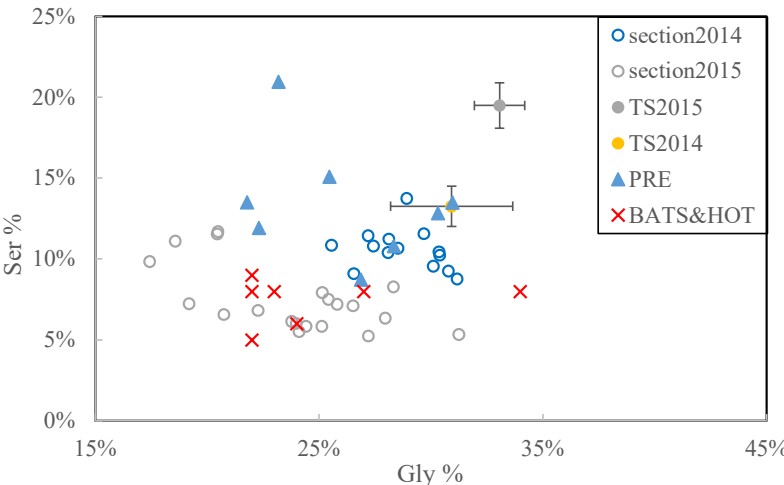

**Figure 6.** Amino acid proportions in the northern SCS and other oligotrophic open sea sites (HOT and BATS). PRE: Peal River Estuary DOM data (salinity > 32) are from Li et al. (2018) [39]; HOT and BATS data are from Kaiser and Benner (2009) [21]. October 2014 and June 2015: this study. Note that the time-series observations are shown as pooled dots with error bar (stdev).

D forms of AAs have been widely used as an indicator of bacterial contribution to organic matter [38,41–43]. The DOM samples had higher D/L Ala ratios in October than in June (Tables 4 and 5), which is consistent with the findings that bacteria were more abundant in October than in June (Table 3). As aforementioned, serine is contained in both diatom frustules (as the L-form) and bacteria membrane detritus (as D-Ser). For the TS2014 and TS2015, which showed contrasting Ser% values (Figure 6), the Ser D- and L-forms were $8.6 \pm 3.3$ nM ($5.2 \pm 1.5$ nM) and $34 \pm 9.3$ nM ($84 \pm 20$ nM) in TS2014 (TS2015), respectively. The D/L-Ser ratio in October (TS2014) was $0.25 \pm 0.06$, much higher than that in June (TS2015; $0.06 \pm 0.009$). Differences between the concentrations of Ser enantiomers, as well as the corresponding D/L ratios, suggest a stronger contribution from phytoplankton (e.g., diatom frustules) than from bacteria at TS2015 than at TS2014, which is consistent with the seasonal bacterial and phytoplankton results (Table 3). We should also bear in mind that the serine enrichment in phytoplankton detritus [40,41] is based on a particulate approach study. Organic matter changes during particulate-dissolved phase transformation and its impact on the serine enrichment pattern are unclear and need further study.

In the oceans, the presence and enrichment of non-protein AA GABA indicates the aging and advanced diagenesis status of the DOM [44]. Moreover, due to the contribution from bacteria in DOM diagenesis, the D/L ratio of DOM is also expected to increase along with the early diagenesis progress. Though with an outlier, of all the time-series and section samples, the relation between GABA% and D/L Ala is statistically negative

(r = 0.32, *p* = 0.005, n = 75). When GABA% was plotted against the D/L Ala ratio, samples from TS2015 showed the lowest GABA% and D/L Ala ratio relative to all other samples, being located in the left bottom corner (Figure 7). The lower GABA% and higher D/L Ala ratio of TS2015 is consistent with its corresponding higher Ser% content (Figure 6). Both figures (Figures 6 and 7) support the idea that the DOM of samples from TS2015 was likely affected by phytoplankton instead of by bacterial rework.

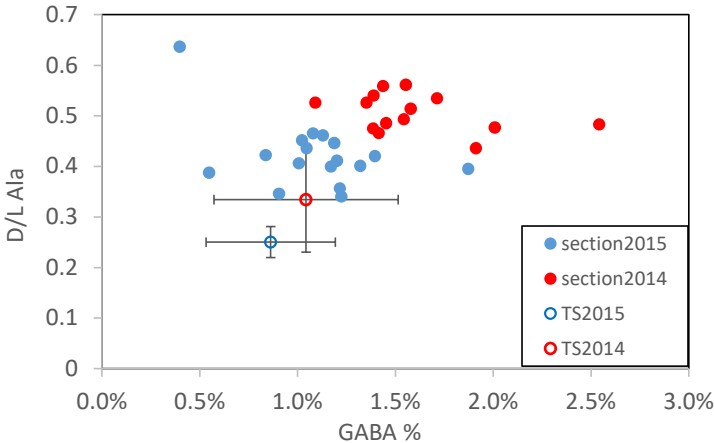

**Figure 7.** Gly% plotted against D/L Ala ratio in the northern SCS slope region. Note that the time-series observations are shown as pooled dots with error bar (stdev).

When comparing both section and time-series observations, DOM in surface waters in June 2015 showed less GABA% and D/L Ala ratio when compared to that in October, 2014 (Figure 7), suggesting that DOM in June 2015 was overall more fresh (less degraded) when compared to that in October 2014. We note that such seasonal DOM degradation pattern is consistent with the seasonal bacterial and phytoplankton biomass pattern (Table 3). The contribution of the eddy-pair in June during our observation is highly suspicious. In June, our section and time-series observations were under a mild upwelling effect, which can be judged from both the sea level anomaly (Figure 1) and the section density profiles (Figure 4). Accordingly, higher chlorophyll was found in June relative to October (Table 3). Though the upwelling of deeper waters also brings up highly degraded DOM from the deep, the upwelling-promoted algae growth and the addition of newly assimilated organic matter into the surface DOM pool seem to play a more important role in determining the overall apparent DOM degradation status in the upper ocean.

## 5. Conclusion and Perspective

The THDAAs and their related molecular indicators reveal a bulk DOM diagenesis status that can correspond to thousands of years, e.g., [21]. Against this background, short-term changes in THDAAs and AA-based molecular indicators, if they exist, would interfere with our understanding of long-term carbon cycling. Hence, the study of THDAAs and their molecular indicators' short-term changes has a potential significance on long-time scale materials cycling.

Overall, our work indicates that care should be taken when using THDAAs and their related molecular indicators to determine the DOM's degradation status. This is because some THDAA molecular indicators showed solid short-term changes within 1 day. The scale of variability within 1 day can be viewed as a resolution of the corresponding molecular indicators when elucidating the DOM composition changes over longer time scales. Such molecular indicators include the AA carbon yield and D/L Ala, which show 0.3–0.4% and 0.02–0.13 short-term changes, respectively.

Given these revealed short-term changes, we applied our findings to DOM seasonal composition comparison. We conclude that DOM in June 2015 is more fresh when compared to that in October 2014. This is quantified by the higher Ser% and lower GABA%

and D/L Ala in DOM's composition in June relative to October. The seasonal differences of these molecular indicators are larger than their corresponding short-term changes. The reason for such DOM composition seasonality is the fact that there was an eddy-pair event going on during our observation in June 2015. Accordingly, more upwelled deeper waters, better nutrient concentration, and higher phytoplankton standing stock are reported in June relative to October. Our finding of such seasonal DOM composition differences, confirms the seasonal DOM's lability difference, which sheds light on the microbiol food loop and marine carbon cycling studies in the SCS.

**Supplementary Materials:** The following are available online at https://www.mdpi.com/2073-4441/13/5/685/s1; Analytical solutions for calculating the propagated errors in molecular indicators (1); Errors in primary measurements and propagated errors in molecular indicators (2), Table S1: Statistical parameters used for calculating the propagated error in DI in this study; Table S2: Errors (standard deviation) and variance in the primary measurements in this study; Table S3: Propagated errors (as transferred standard deviation, in bold) in the molecular indicators in this study.

**Author Contributions:** Conceptualization: Z.-Y.Z.; data curation, Y.-C.Z., W.-C.M., Y.W., M.L., S.-M.L., and X.-W.X.; investigation, Z.-Y.Z.; methodology, M.L.; resources, Y.W., S.-M.L., X.-W.X., and M.Z.; visualization, W.-C.M.; writing—original draft, Z.-Y.Z.; writing—review & editing, Y.-C.Z. All authors have read and agreed to the published version of the manuscript.

**Funding:** This research was funded by the Ministry of Science and Technology of China through a '973' project (no. 2014CB441503 and 2014CB441502), and by National Natural Science Foundation of China (no. 41676188).

**Institutional Review Board Statement:** Not applicable.

**Informed Consent Statement:** Not applicable.

**Data Availability Statement:** The data presented in this study are available in supplementary material.

**Acknowledgments:** We are grateful to the captain and crew of R/V *Nanfeng*, and to the students from SKLEC/ECNU and SOO/SJTU who helped us with the field work and sample collection. We are especially grateful for the help of Jun Xu and Yu Zhang from SOO/SJTU. Lü Ni from DSAS/ECNU participated in the statistical inference. Zhuoyi Zhu would like to thank Zhihao Zhang and Shipan Hossen for their assistance with the figure plotting. This work was financially supported by the Ministry of Science and Technology of China through a "9732" project (no. 2014CB441503 and 2014CB441502), and by the National Natural Science Foundation of China (no. 41676188).

**Conflicts of Interest:** The authors declare no conflict of interest.

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
