# Peer review of "Characterization of Marine-Surface-Dissolved Organic Matter via Amino Acid Enantiomers and Its Implications Based on Diel and Seasonal Observations"

_water, doi:10.3390/w13050685_

Round 1
Reviewer 1 Report
The Authors present a research work on a limnetic subject, of total interest in opinion of this reviewer. The research focuses on the phytoplankton biomass quality & evolution, and its characterization through amino acids and marine dissolved organic matter in their role of tracers.
The study is relevant and deserves publication. This reviewer has some experience on limnetic field works to assess the quality of water bodies, being familiar with all the unavoidable paraphernalia involved in those studies.
The Introduction is OK and provides the reader with a reasonably complete background on the subject. The general writing pose of the document is correct, with adequate structure and English. The figures are fine in general, although in some of them, specially the contour maps, the font size of isolines and axis labels should be increased to ease the reading, and the general quality of inserted images should be improved. The length of the paper is simply perfect.
The Authors might want to take a look back at the Abstract in order to: 1) define de acronym DOM when it appears first and not in the second line (once it has been introduced first), 2) try to simplify the profusion of data towards the end of it, and comment some relevant yet useful outcome of the research instead...Which leads this reviewer to what it is the main issue (if it can be dubbed so) of the paper. While the research subject is fine, the Introduction and Conclusion sections fail somehow in the following terms:
- The Introduction should provide with a clear scope of what is the gap bridged by this research. To put it bluntly, the Authors might want to clearly state whether their research is a field work based study strengthening some limnetic facts already known/studied by others, or otherwise they come up with relevant results to be used in the next future.
- The Conclusion seems to be more a repetition of the Results and Discussion sections, rather than a closure of the research in which some outstanding findings are revealed. In fact, the last phrase "Due to the essential roles of DOM in both microbiol food loop and marine carbon cycling? changes in surface water DOM lability ?freshness? and its impact to marine ecosystem and carbon cycling needs further study?" might be a little disappointing, in a sense that the results of all the research and field works should be more relevant that a perspective of a "future study" with no more information revealed, so to speak. This reviewer would suggest the Authors that, aside form the fact of the open line and future research, they might want to reconsider the Conclusion and try to draw some general patterns highlighting how AA and DOM tracing reveals natural (or not) processes that in turn could affect other links in the trophic chain. Or if they could be used to reveal any other patterns (even climate change associated processes if the Authors consider it appropriate).
Reviewer 2 Report
Line 19: check the use of the term ”respectively”.
Line 113, figure 1: Scale on the right should have units.
Line 113, figure 1: Scale to the right in meters as indicated in the figure caption? Table 2 shows 200 m of depth!
Line 183: Do not separate the size of the units (10 L).
Line 228, table 2: Add a space between the quantities and the units. It must be 200 m and not 200m.
Line 228, table 2: Values below the detection limit should be referred as (nd) and not zero. For these values, the statistical parameters should not be indicated.
Line 252: In this paragraph, and the rest of the text, AA abbreviations must not have the suffix %. The % is referred in the values of the results.
Line 291, figure 2: These figures does not have sufficient quality for publication. Its reading is difficult.
Line 296: “Waters with density 22.5 kg / m3” seawater has a density of approximately 1025 kg / m3. It is noticed that the values above 1000 are used but this nomenclature widely used must be clarified and indicated in its first use.
Line 296: In writing m3, 3 must be superscript.
Line 324, figure 4, The font size on the x and y axes and inside the figures is not legible.
Line 333 and following: The term GABA must be used without %. Units must be associated with numerical quantities, not their abbreviations.
Line 362:Do not separate quantities (24) and units (h) in different lines.
Line 375: r = - 0.55, write on a single line.
Line 407, figure 6: Remove border from the figure. Axis titles (Gly and Ser) should not have the prefix %. The scale of the graph has the units. Alternative: put in the titles of the axes Gly (%) and Ser (%) and remove the % from the scale numbers.
Line 407, figure 6: Improve the quality of the figure.
Line 430, figure 7: The same observations for the x axis as for figure 6.
Line 430, figure 7: The numbers on the x scale have an extra zero on the right.
Each reference should have the first line indented ~1 cm to the left.
Round 2
Reviewer 1 Report
This reviewer thanks the Authors for their gentle answer to all the suggestions.
Author Response
Dear reviewer,
I have rechecked the manuscript for its language issue. I correct a few words that is mis-spelled.
I think now it is the best I can do for the manuscript.
If I missed any comments or suggestions, please let me know.
Thank you
Zhuoyi